# Layerwise Importance Analysis of Feed-Forward Networks in Transformer-based Language Models

**Wataru Ikeda**[α]**, Kazuki Yano**[α]**, Ryosuke Takahashi**[α]**, Jaesung Lee**[α]**,
Keigo Shibata**[α]**, & Jun Suzuki**[αβγ]
[α]Tohoku University,   [β]RIKEN,   [γ]NII LLMC
ikeda.wataru@dc.tohoku.ac.jp

## Abstract

This study investigates the layerwise importance of feed-forward networks (FFNs) in Transformer-based language models during pretraining. We introduce an experimental approach that, while maintaining the total parameter count, increases the FFN dimensions in some layers and completely removes the FFNs from other layers. Furthermore, since our focus is on the importance of FFNs during pretraining, we train models from scratch to examine whether the importance of FFNs varies depending on their layer positions, rather than using publicly available pretrained models as is frequently done. Through comprehensive evaluations of models with varying sizes (285M, 570M, and 1.2B parameters) and layer counts (12, 24, and 40 layers), we demonstrate that concentrating FFNs in 70% of the consecutive middle layers consistently outperforms standard configurations for multiple downstream tasks.

## 1 Introduction

Language models based on Transformer architectures (Vaswani, 2017) have rapidly evolved and are now a central research topic in the fields of natural-language processing and artificial intelligence. In such Transformer-based language models (hereinafter, "Transformer LMs"), many detailed model designs have been proposed and implemented. Conceptually, most Transformer LMs comprise stacked Transformer layers, each containing two main components: a self-attention mechanism and a feed-forward network (FFN) (Vaswani, 2017; Touvron et al., 2023). Figure 1(a) illustrates a standard Transformer layer. The computational process follows a specific pattern, particularly in the Transformer LMs with the pre-language-normalization (pre-LN) (Xiong et al., 2020a). Each layer sequentially processes the input vectors through both self-attention and FFN components, and the resulting output vectors are added to the original token embeddings through residual connections. This process is repeated across all layers to progressively refine the representation and ultimately produce the final hidden-state vectors.

Numerous previous studies have attempted to investigate the roles of the self-attention and FFN components, both individually and simultaneously, to understand what Transformer LMs compute internally. Most studies conclude that the self-attention mechanism mainly handles the mixing of information obtained from token embeddings, while the FFN primarily serves to store knowledge from the training data (Geva et al., 2021; Dai et al., 2022; Meng et al., 2022).[1] Assuming that FFN layers embed knowledge, functioning similarly to a key-value memory, many questions remain, such as whether FFNs are the most effective form for acquiring knowledge and where exactly within the multiple layers of a Transformer this knowledge is embedded.

To better understand the role of FFNs within Transformer LMs, we use an original approach to uncover some of their roles and functions. More specifically, instead of evaluating publicly

---

[1]Although some empirical and theoretical studies (e.g., Kobayashi et al. (2024)) have demonstrated that FFNs may include other functions and effects, they do not prove that FFNs do not store knowledge.

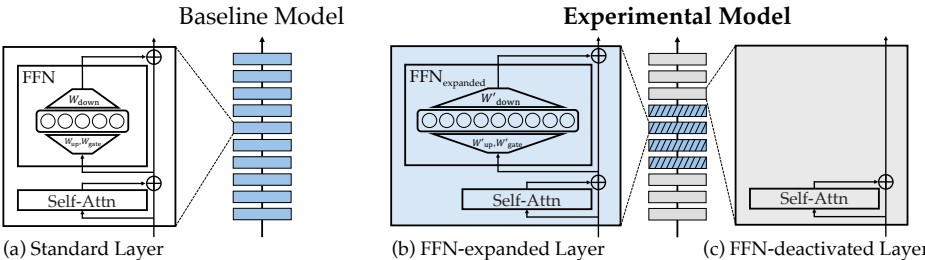

Figure 1: **Layer Structure of the Baseline Model and Our Experimental Model.** (a) Standard Transformer layer: The baseline model is a standard stack of Transformer layers as implemented in LLaMA. (b) FFN-expanded layer: In the experimental model, certain layers have an expanded intermediate representation dimension in the FFN. (c) FFN-deactivated layer: In the remaining layers, the FFN is removed. In our experimental model, while maintaining the overall parameter count of the baseline model, the FFN's computational capacity (i.e., the number of parameters) is concentrated in specific layers.

available (static) pretrained models, we evaluate Transformer LMs trained from scratch with several nonstandard FFN configurations within the Transformer layers, such as models with removed or enlarged FFNs. By comparing the task performance of such nonstandard FFN configurations, we aim to clarify whether the importance of FFNs depends on their positions within the Transformer LMs during the pretraining phase.

## 2 Related Works

Several studies have demonstrated that FFNs store knowledge and that specific neurons play an important role in representing and recalling factual information (Geva et al., 2021; Dai et al., 2022; Meng et al., 2022). Additionally, Kobayashi et al. (2024) proposed a different interpretation of the role of FFNs, showing that FFNs, together with layer normalization, contribute to contextualizing inputs.

More recently, studies have analyzed LMs at the layer level and have reported that mid-depth layers provide robust representations, whereas the final layers tend to overspecialize toward the pretraining objective (Skean et al., 2025).

While many important insights have emerged, these findings derive from analyses of pretrained LMs and are limited to standard model architectures. The verification method used in the present study is original because it examines the impact of modifying the model architecture itself, such as removing FFNs from specific Transformer layers or increasing the dimension of the remaining FFNs. Adopting such a structural approach to model investigation rather than conventional analytical methods should lead to new insights into the functions and roles of FFNs.

## 3 Model Settings for Identifying the Position-based FFN Importance

This study examines the importance of FFNs as a function of their layer position within a Transformer LM. To focus our investigation, we limit this study to the LLaMA architecture (Touvron et al., 2023), which has become the *de facto* standard model for Transformer LMs. Specifically, LLaMA integrates modern architectural improvements, including pre-LN (Xiong et al., 2020b), SwiGLU activation function (Shazeer, 2020), and RoPE positional encoding.

Based on this model architecture, a Transformer layer, which in this paper refers to each layer in a Transformer LM, primarily consists of a self-attention mechanism followed by an FFN. Although the Transformer layer includes layer normalizations with a pre-LN setting,

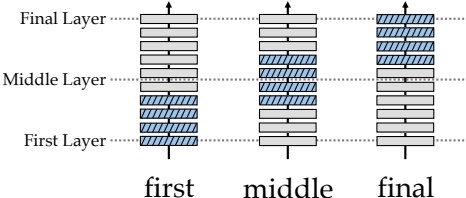
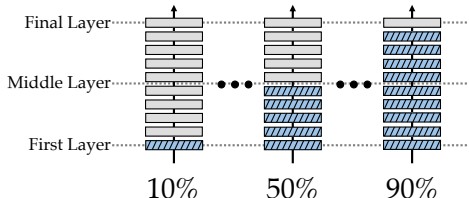

Figure 2: **Different Positional Configurations of the FFN-extended Layer.** In our experimental model, the FFN-expanded layer is placed in one of three positions: near the input layer (first), in the middle layers (middle), or near the output layer (final) and we evaluate the effects of these different placements.

Figure 3: **Different Placement Ratios of the FFN-expanded Layer.** In our experimental model, we vary the proportion of FFN-expanded layers and evaluate their effects. For instance, when placing FFN-expanded layers near the input layers (first), we define what percentage of all layers they represent.

we omit explicit explanations about layer normalization in our subsequent discussions because these details are not critical to our investigation and analysis.

### 3.1 Transformer Layers Characterized by Different FFN Types

Figure 1 shows the three types of Transformer layers used in this study: **standard**, **FFN-expanded**, and **FFN-deactivated**. These are characterized only by the FFN settings, which are explained below in detail.

**Standard Layer (baseline models).** The standard layer refers to the standard setting of the Transformer layer in LLaMA and serves as the baseline setting. The LLaMA uses the SwiGLU activation function. Let $\sigma(\cdot)$ be the element-wise sigmoid function whose input is a vector. The FFN then takes an input vector $x \in \mathbb{R}^d$, where $d$ is the dimension of the hidden input and output vectors, and processes it by expanding it to an intermediate representation dimension $d_f$ through projection matrices $W_{\text{gate}}$ and $W_{\text{up}} \in \mathbb{R}^{d_f \times d}$, followed by a projection back to the original dimension via $W_{\text{down}} \in \mathbb{R}^{d \times d_f}$:

$$\text{FFN}(x) = W_{\text{down}}(\text{Swish}(W_{\text{gate}}x) \otimes W_{\text{up}}x), \quad \text{where} \quad \text{Swish}(x) = x\sigma(x). \tag{1}$$

**FFN-expanded and FFN-deactivated layer (experimental models).** We define the FFN-expanded layer as essentially identical to the standard layer, except that the intermediate representation dimension of the FFN, $d_f'$, is expanded [see Figure 1(b)]. In other words, the relation $d_f < d_f'$ holds. We define the FFN-expanded layer $\text{FFN}_{\text{expanded}}$ as

$$\text{FFN}_{\text{expanded}}(x) = W_{\text{down}}'(\text{Swish}(W_{\text{gate}}'x) \times W_{\text{up}}'x), \tag{2}$$

where $W_{\text{gate}}', W_{\text{up}}' \in \mathbb{R}^{d_f' \times d}$ and $W_{\text{down}}' \in \mathbb{R}^{d \times d_f'}$. Section 4.2 explains how we determine $d_f'$.

We define the FFN-deactivated layer as the standard LLaMA layer, with the FFN removed entirely from the Transformer layer. Note that this approach means that the FFN-deactivated layer consists only of the self-attention mechanism [see Figure 1(c)].

### 3.2 Layer Placements within the Transformer LMs

Using the three types of Transformer layers explained in Section 3.1, we define three types of Transformer layer placements: {first, middle, final}. Conceptually, each label indicates the approximate relative position of the FFN-expanded layers. Figure 2 illustrates this configuration. More specifically, first means that we assign FFN-expanded layers to the first (next to the input) up to a specified percentage of subsequent layers and assign FFN-deactivated layers to the remaining layers. Similarly, final assigns FFN-expanded layers

starting from the final (just before the output) through a specified percentage of preceding layers and assigns FFN-deactivated layers to the remaining layers. Finally, `middle` assigns a specified percentage of the FFN-expanded layers symmetrically about middle layer at $L/2$ and assigns the FFN-deactivated layers to the remaining positions. These three layer placements (`first`, `middle`, `final`) essentially redistribute the computational capacity and parameters within the model, removing the FFNs from some layers while expanding them in others. This redistribution can be viewed as concentrating the FFN's representational capacity in specific layers while maintaining the same overall parameter budget.

The design of our experiment was motivated by the goal of structurally verifying during pretraining the hypothesis that FFNs store knowledge. Specifically, our primary operation of "removing FFNs" is intended to examine whether FFNs in other layers may serve as alternative storage for the knowledge that would normally be stored by FFNs in specific layers under standard uniform FFN placement (in situations where FFNs in those layers are absent). Furthermore, by combining this approach with the "expansion operation" that redistributes the parameters lost through removal to FFNs in specific layers, we investigate whether knowledge accumulation can be concentrated.

By strategically placing these modified layers, we can investigate whether certain positions within the network benefit more from enhanced FFN capacity than others—that is, which layer positions allow FFNs to effectively store knowledge. By maintaining the total number of parameters in all experimental models, we can measure the effects of placement and systematically verify the layerwise nature of the FFNs' knowledge-storage function.

## 4 Experiments

This section explains the experiments we conducted in this paper. Using the standard pretraining procedure, we trained the Transformer LMs from scratch using different model sizes and layer counts with baseline, `first`, `middle`, and `final` settings. Next, we evaluated the pretrained models based on the standard benchmark datasets, which are often used to assess Transformer LMs.

A performance degradation for certain model configurations indicates that the positions in the model of the FFN-deactivated layers are important for maintaining performance. Using this approach, we investigate whether removing FFNs from certain layers degrades or improves performance, thus revealing the importance of layers to the function of FFNs.

### 4.1 Baseline Model Setup

We constructed three baseline models with different numbers of parameters and layer configurations to ensure that the results are robust across varying model architectures. Our baseline configurations include (1) a 285M parameter model with 12 layers, a hidden dimension $d$ of 1280, and an FFN intermediate dimension $d_\mathrm{f}$ of 4480; (2) a 570M parameter model with 24 layers, maintaining the same hidden and intermediate dimension sizes; and (3) a 570M parameter model with 40 layers, using a smaller hidden dimension size of 992 and an FFN intermediate dimension size of 3472.

The rationale behind these diverse baseline configurations is twofold: First, comparing the 285M and 570M models allows us to detect whether any trends regarding FFN importance are independent of model sizes. Second, comparing the 24-layer variant with the 40-layer variant of the 570M model enables us to examine whether any observed patterns are independent of the number of layers, which is particularly important because our approach removes FFNs from a certain percentage of layers. Given that major models such as LLaMA 8B have 32 layers (Grattafiori et al., 2024) and Qwen3 14B has 40 layers (Yang et al., 2025), this 40-layer configuration covers a practical range of the layer count. This multiconfiguration approach helps us explore the consistency of the behavioral patterns across different model architectures.[2]

---

[2]Appendix A provides detailed configurations of these baseline models.

## 4.2 Experimental Model Setup

The experimental models maintain the same basic configuration as the baseline models, including the number of layers and hidden dimensions. In the experimental models, we replace the baseline model's layers with either FFN-expanded layers or FFN-deactivated layers according to the layer placement positions described in Section 3.2 and the ratio of FFN-expanded layers described below.

**Ratio $r$% of FFN-expanded layers to total layers, where $r \in \{10, 30, 50, 70, 90, 100\}$.** As illustrated in Figure 3, the total number of FFN-expanded layers is determined by the product of the total number $L$ of layers and $r$, rounded down to the nearest integer ($\lfloor rL/100 \rfloor$).

Combining the placement positions from the previous subsection with these ratio configurations, we generate experimental models for each baseline model. For example, in the 285M, 12-layer baseline model with $r = 30$% and the `middle` position, we place FFN-expanded layers in layers 6, 7, and 8 (because $\lfloor 12 \times 0.3 \rfloor = 3$ layers), with the remaining layers being FFN-deactivated.[3]

Importantly, all experimental models maintain the same total parameter count as their corresponding baseline models. This parameter parity is achieved by expanding the intermediate dimension size $d_f'$ of FFN-expanded layers to compensate for the parameters removed from the FFN-deactivated layers. The intermediate dimension $d_f'$ of FFN-expanded layers is recalculated based on the ratio of the FFN-expanded layers in each experimental model configuration and determined such that the total parameter count remains nearly identical to the baseline model.[4]

By combining the two factors above, we established 18 different experimental configurations (six ratios of FFN-expanded layers times three positions) for each baseline model (285M, 12 layers; 570M, 24 layers; and 570M, 40 layers). Note that, when the ratio of FFN-expanded layers is 100%, the model architecture is identical to the baseline architecture, so we simply use the baseline results rather than training a redundant model for these configurations.

## 4.3 Pretraining and Evaluation

**Pretraining and Evaluation.** For pretraining the baseline and experimental models, we used standard pretraining methods with the FineWeb-Edu dataset (Lozhkov et al., 2024).[5] We evaluated the pretrained models in terms of the downstream task performance and knowledge capacity. For downstream task evaluation, we used the `lm-evaluation-harness` framework (Gao et al., 2024) with a diverse set of tasks: LAMBADA (Paperno et al., 2016) for contextual next-word prediction, Wikitext (Merity et al., 2017) for language modeling, Winogrande[6] (Sakaguchi et al., 2020) and PIQA (Bisk et al., 2020) for commonsense reasoning in a binary choice format, HellaSwag (Zellers et al., 2019) for selecting the most natural continuation of a context, and ARC[7] (Clark et al., 2018) for scientific knowledge and reasoning. We used accuracy (Acc) as the evaluation metric for the choice-based tasks (ARC, HellaSwag, PIQA, and Winogrande), and we evaluated both accuracy (Acc) and perplexity (PPL) for LAMBADA. Finally, we used perplexity (PPL) for Wikitext.

**Knowledge Capacity Evaluation.** Prior research hypothesizes that FFNs serve as knowledge storage components in Transformer LMs (Geva et al., 2021; Dai et al., 2022; Meng et al., 2022), so it is natural to investigate how architectural modifications to FFNs might affect the amount of knowledge stored in a model. To quantitatively evaluate this aspect, we use the Zero-Shot Relation Extraction (zsRE) dataset (Levy et al., 2017) to measure knowledge

---

[3]For the `middle` configuration with an odd number of FFN-expanded layers, we place one more layer in the latter half of the model with respect to the middle layer ($L/2$).

[4]Appendix B shows the specific intermediate dimension sizes $d_f'$ for each experimental model.

[5]Detailed pretraining configurations are provided in Appendix C.

[6]Hereinafter, we refer to Winogrande as WinoG.

[7]ARC consists of two subsets: the Easy set and the Challenge set, which we refer to as ARC-e and ARC-c, respectively.

capacity, following the methodology of Mitchell et al. (2022); Cao et al. (2021). The dataset consists of knowledge-based question-answer pairs, allowing us to evaluate the model's ability to retrieve factual information.[8]

**Performance Metrics.**   For each evaluation result, we calculate the **relative improvement (RI)** with respect to the baseline to facilitate comparison between the experimental models and the baseline. This calculation is done as follows:

$$\text{RI}(m, T) = s(T) \times \frac{\text{metric}(m, T) - \text{metric}(\text{baseline}, T)}{\text{metric}(\text{baseline}, T)} \times 100 [\%] \tag{3}$$

where $s(T)$ is a sign-correction factor defined as

$$s(T) = \begin{cases} 1 & \text{if task } T \text{ uses accuracy-based metrics} \\ -1 & \text{if task } T \text{ uses loss-based metrics (e.g., perplexity)} \end{cases} \tag{4}$$

Here, $\text{metric}(m, T)$ gives the metric of experimental model $m$ applied to task $T$, and $\text{metric}(\text{baseline}, T)$ gives the metric of the baseline model applied to task $T$. The sign-correction factor $s(T)$ ensures that a positive RI consistently indicates better performance regardless of whether the underlying metric follows a "higher-is-better" convention (e.g., accuracy) or a "lower-is-better" convention (e.g., perplexity). Zero RI indicates that the performance is equivalent to that of the baseline model, positive (negative) RI indicates that the performance exceeds (is inferior to) that of the baseline.

## 5   Results

Figure 4 compares the RI of various FFN configurations with the RI of the baseline.[9]  To ensure a fair comparison, we excluded the results of certain downstream tasks when either the baseline model or experimental model performed below the chance level because such results would not provide meaningful insights into architectural differences.

Hereinafter, we refer to individual results using the following format "[pos]_[pct]_[size] _[layers]", where [pos] is the position of the FFN-expanded layers (first, middle, or final), [pct] is the percentage of the FFN-expanded layers, [size] is the model size (e.g., 285M), and [layers] is the total number of layers. For example, middle_30_285M_12l refers to a model with 285M parameters and 12 layers, where 30% of the layers are FFN-expanded layers positioned in the middle of the architecture.

In this section, we analyze these results with respect to (1) the ratio of the FFN-expanded layers and (2) their position within the model.

### 5.1   Effectiveness of FFN Expanded Layer Ratio

Our results reveal a clear relationship between the ratio of the FFN-expanded layers and the model performance. Models with low layer ratios (10%–30%) consistently performed worse than the baseline model in nearly all evaluations, and the performance degradation is substantial for numerous tasks: HellaSwag produces a relative degradation ranging from $-0.35\%$ to $-6.53\%$ excluding final_30_285M_12l [Figure 4(b)], and Wikitext perplexity produces a relative degradation from $-19.07\%$ to $-1.57\%$ for all model sizes [Figure 4(a)]. Although LAMBADA accuracy and zsRE improve the performance with respect to the baseline model in the 570M_40l configuration, the overall trend remains negative across most experimental settings [Figures 4(c) and 4(d)].

As the FFN-expanded layer ratio increases, performance improves for almost all tasks. Ratios of 70%–90% produce a consistent trend whereby increasing the configuration produces

---

[8]Appendix D provides detailed procedures for this knowledge assessment.
[9]Additional evaluation results for tasks not shown in Figure 4 are provided in Appendix F. Moreover, to validate the meaningfulness of our RI-based analysis, we show the absolute metric of our baseline models with metrics of models from the literature in Appendix E.

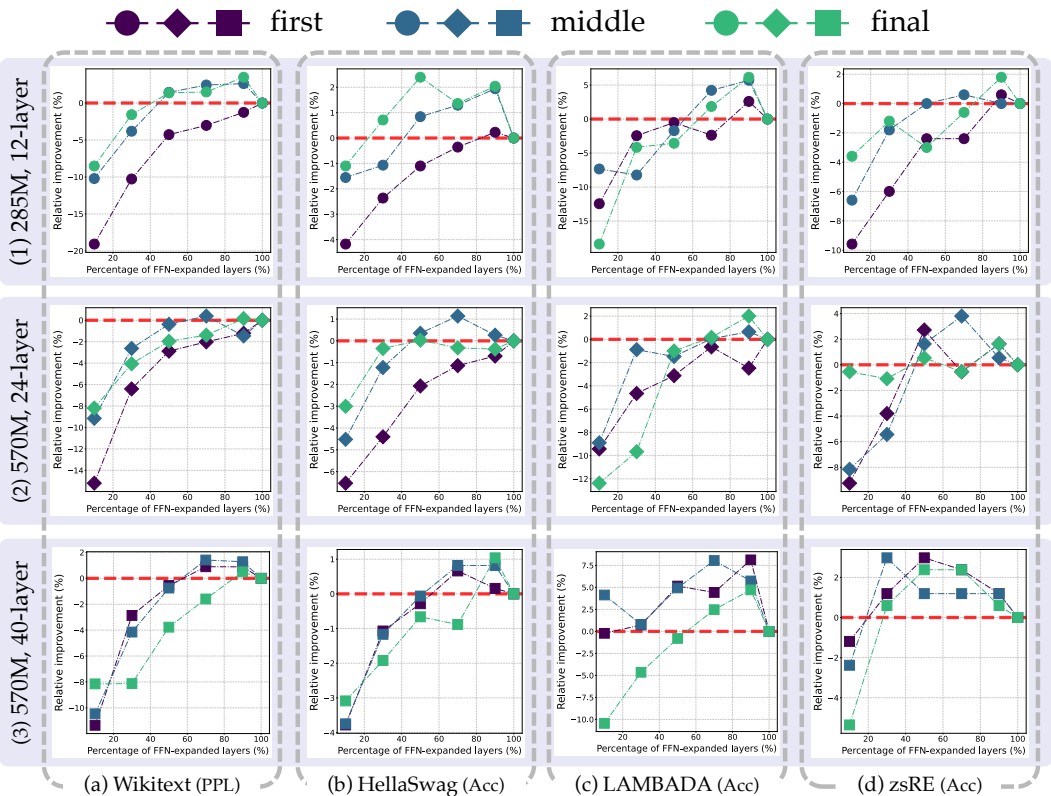

Figure 4: **Relative Improvement across Tasks by FFN-expanded Layer Ratio.** Relative improvement across tasks as a function of FFN-expanded layer ratio for different placement positions. Each row represents a different baseline configuration (model size and number of layers), while each column shows results for a different evaluation task. In each graph, the red dashed line highlights zero relative improvement, representing performance equivalent to the baseline model. Note that at 100% ratio, all configurations converge to the baseline performance regardless of placement position.

a model that outperforms the baseline model, although the gains vary by task configuration.[10] Particularly for the 285M_12l and 570M_40l models, most experimental models within this range of FFN-expanded layer ratio outperformed the baseline model for all tasks [Figures 4(1) and 4(3)]. These findings suggest that extreme concentration of FFN parameters in very few layers compromises model capability, likely because reducing the layers applying nonlinear transformations limits the representational capacity, even with individually larger FFNs.

## 5.2 Effectiveness of FFN-expanded Layer Position

Focusing on the 70%–90% FFN-expanded layer ratio range where the models tend to outperform the baseline model, we observe notable differences based on the position. The middle and final configurations consistently outperform the first configuration across most tasks in the 285M (12-layer) and 570M (24-layer) models.

For the 285M model, Figure 4(1)(b) shows that, at 90% ratio, the first position yields only a +0.23% improvement with respect to HellaSwag, whereas the middle and final positions yield +1.94% and +2.04%, respectively. This pattern repeats for the LAMBADA accuracy

---

[10]Task performance gains naturally vary due to differences in task difficulty. We address this challenge by examining the consistency of performance for multiple evaluation tasks and model configurations with different sizes and layer counts.

| 285M (12-layer) | | 570M (24-layer) | | 570M (40-layer) | |
|---|---|---|---|---|---|
| Model | Avg RI (%) | Model | Avg RI (%) | Model | Avg RI (%) |
| final_90 | +4.72 | **middle_70** | +1.81 | first_90 | +3.41 |
| middle_90 | +4.35 | final_90 | +1.34 | **middle_70** | +3.36 |
| **middle_70** | +3.78 | middle_50 | +0.70 | middle_90 | +2.62 |
| final_70 | +1.94 | middle_90 | +0.26 | first_70 | +2.35 |
| middle_50 | +1.37 | final_70 | +0.12 | first_50 | +2.15 |

Table 1: **Average Relative Improvement (Avg RI) for Top 5 Models by Model Size.** Avg RI shows the mean value across six downstream tasks (Wikitext, LAMBADA, HellaSwag, zsRE, ARC-e, PIQA).

| Model | Wikitext PPL | LAMBADA PPL | LAMBADA Acc | ARC-e Acc | ARC-c Acc | WinoG Acc | PIQA Acc | HellaSwag Acc | zsRE Acc | Avg |
|---|---|---|---|---|---|---|---|---|---|---|
| first_70 | +0.30 | −0.30 | +2.21 | **+1.89** | −0.60 | **+0.61** | +0.71 | −0.64 | +2.04 | +0.69 |
| middle_70 | **+1.09** | **+6.47** | **+2.89** | +1.55 | −3.60 | −1.83 | **+1.27** | **+0.72** | **+3.06** | **+1.29** |
| final_70 | −1.71 | +1.55 | +0.74 | −0.54 | **+3.90** | −1.22 | +0.64 | −1.25 | +2.55 | +0.52 |

Table 2: **Relative Improvement (%) of Experimental Models over Baseline for 1.2B Parameter Models.**

[Figure 4(1)(c)] and Wikitext [Figure 4(1)(a)], where first_90 increases RI by +2.59% and −1.26% compared whereas final_90 increases RI by +6.15% and +3.49%, respectively.

The 570M, 24-layer model produces similar trends, with middle_70 achieving +1.12% on HellaSwag, whereas first_70 achieves only −1.14% [Figure 4(2)(b)]. In the knowledge assessment through zsRE [Figure 4(2)(d)], middle_70 produces a remarkable +3.80% improvement, significantly outperforming first_70, which finishes at −0.54%.

Curiously, this pattern becomes less consistent in our larger 40-layer experiments. As shown in Figure 4(3), the final configuration occasionally underperforms both the middle and first configurations. For example, final_90 achieves a +4.72% improvement, whereas first_90 achieves +8.13% in the LAMBADA accuracy [Figure 4(3)(c)].

These findings suggest that FFNs positioned in the middle to later layers contribute more to model performance than FFNs in earlier layers, particularly in models with moderate layer counts (12–24).

## 5.3 Top 5 Experimental Models

Although our analysis reveals general trends across different FFN-expanded layer positions, identifying specific high-performance configurations is crucial. Therefore, we calculated the average RI for all downstream tasks for each experimental model. Table 1 presents the top five configurations based on the average RI for each model size. Notably, the middle_70 configuration consistently performs well for all model scales, ranking third (+3.78%) for the 285M model, first for the 570M, 24-layer (+1.81%) model, and second for the 570M, 40-layer (+3.36%) model. This consistency suggests that concentrating FFNs in approximately 70% of the layers around the center of the network is a robust architectural choice.

Based on these findings, we identify middle_70 as the most promising configuration and extend our analysis to larger models, including first_70 and final_70 for comparison to further validate the effect of position when scaling.

## 5.4 Scaling to 1.2B Parameter Models

To further validate our findings and determine whether the observed patterns persist at larger scales, we conducted additional experiments with a 1.2B parameter model using

a 40-layer configuration.[11] Table 2 presents the RI of each configuration compared with the baseline RI across the eight downstream tasks. The `middle_70` configuration produces the highest average improvement ($+1.29\%$) for all tasks, outperforming both `first_70` ($+0.69\%$) and `final_70` ($+0.52\%$). This pattern is consistent with our results for both the 285M and 570M models, for which the `middle_70` configuration consistently ranks among the top-performing models.

Considering individual tasks, the `middle_70` configuration excels particularly in language modeling and knowledge-intensive tasks, demonstrating the biggest improvements on Wikitext ($+1.09\%$), LAMBADA (PPL: $+6.47\%$, Acc: $+2.89\%$), and zsRE ($+3.06\%$).

These results from the 1.2B parameter model provide strong evidence that concentrating the FFN parameters in specific layers rather than distributing them uniformly across all layers can significantly improve the downstream task performance. Our experimental approach notably demonstrates that the `middle_70` configuration (i.e., concentrating FFNs in 70% of consecutive middle layers) consistently performs best for model scales from 285M through 570M to 1.2B parameters. This remarkable consistency across different model sizes suggests that the advantage of strategic FFN layer positioning represents a fundamental architectural property of Transformer LMs rather than a scale-dependent phenomenon.

The underlying rationale for these results is consistent with prior research showing that the most significant information processing for downstream tasks occurs primarily in the mid-to-final FFN layers of the model (Meng et al., 2022; Geva et al., 2021). Our `middle_70` configuration effectively concentrates the parameter budget on the parts of the model that matter most for downstream tasks, thereby utilizing the limited parameters more efficiently.

## 6 Layerwise Importance Analysis

To quantify the contribution of FFNs in each layer to the overall model performance and visualize the layerwise importance of FFNs, we developed a layerwise-importance metric derived from the experimental results of Section 5. This metric is based on the idea that performance degraded upon removing a specific layer's FFN, so that layer's FFN must be particularly important to the model's capabilities.

To analyze different configurations of the FFN-expanded layers, we designed our importance metric based on a methodological starting point and a computational procedure. First, given the technical challenge of directly quantifying the individual contribution of each layer within specific configurations, we assume that when a set of FFN-deactivated layers degrade performance in terms of RI, this degradation is spread equally among all deactivated layers in that configuration. Second, for each layer, we sum its importance over all configurations where it was deactivated through a normalized average.

For example, consider layer index 2 in our 570M, 40-layer model in the `final_50` configuration. With this setting, layers 1–20 are FFN-deactivated, and this configuration produces an average RI of $-2.04\%$ over all evaluation tasks. Applying our first assumption, we attribute an importance of $+0.102\%$ to layer 2 (and the other 19 layers) from this configuration because the $-2.04\%$ degradation is distributed equally among the 20 deactivated layers (i.e., $2.04/20 = 0.102$). This process is repeated for all configurations where layer 2 is deactivated, and the results are averaged to obtain the final importance score.

The metric is designed such that higher values indicate greater importance. When FFN deactivation in certain layers leads to larger performance drops in downstream tasks and knowledge assessment, those layers are assigned higher importance scores.[12]

The bar plot in Figure 5 shows the computed importance scores for different layers, where each score has been standardized (zero mean and unit variance). Positive values (shown in blue) indicate layers where FFNs exert an above-average importance on model performance,

---

[11]Detailed model configurations and pretraining configurations are provided in Appendices A and C, respectively.

[12]The detailed mathematical derivation of this metric is provided in Appendix H.

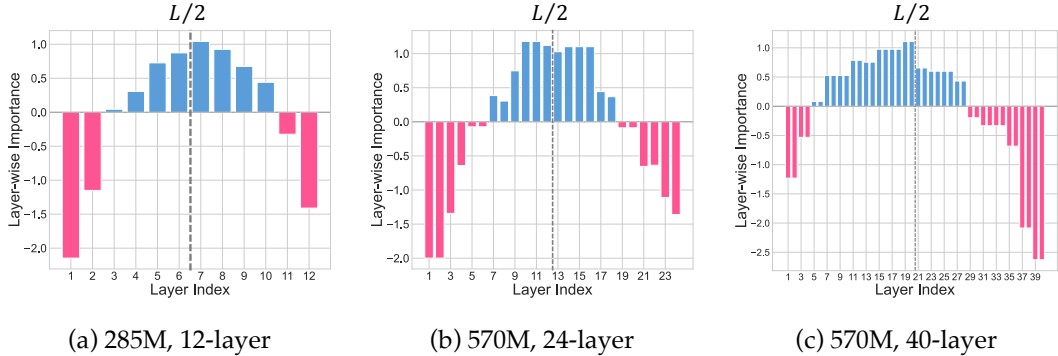

(a) 285M, 12-layer      (b) 570M, 24-layer      (c) 570M, 40-layer

Figure 5: **Layerwise Importance Scores.** The horizontal axis represents the layer index, and the vertical axis represents the corresponding standardized importance score, where higher values indicate that the layer is more important.

whereas negative values (shown in red) indicate layers where FFNs exert below-average importance. The magnitude of each bar reflects the importance of the FFN at the given layer with respect to the importance averaged over all layers.

Analysis of the layerwise importance scores revealed several key patterns across the model scales. First, all three configurations demonstrate a clear concentration of high-importance layers in the middle portion of the networks, while the very first and final layers consistently show below-average importance. In the 12-layer model [Figure 5(a)], layers 3–10 show positive importance scores. Similarly, layers 7–18 of the 24-layer model [Figure 5(b)] are highly important. The 40-layer model [Figure 5(c)] produces positive importance scores for layers 5–28. Second, the distribution of layer importance shifts systematically as the model depth increases. To illustrate this pattern, Figure 5 includes black dotted lines marking the middle position ($L/2$) for each model configuration. Examining the importance distribution relative to this midpoint reveals a clear trend: the 12-layer model concentrates importance somewhat toward the latter half of the network, the 24-layer model produces a more balanced importance distribution about the middle with a slight bias toward the latter half, while the 40-layer model shifts importance toward the earlier portion of the network. This progressive forward shift in the FFN importance distribution, from 12 to 24 to 40 layers, suggests that, as a model deepens, FFNs may become more effective when positioned earlier in the network architecture. This phenomenon might occur because, in deeper networks, hidden states execute more self-attention functions before reaching the middle layers, potentially resulting in overcontextualized representations that FFNs may struggle to process effectively.

## 7   Conclusion

This paper investigates the layerwise importance of FFNs, one of the component elements of Transformer LMs, focusing on their position-dependent significance within the overall model architecture during the pretraining process. By evaluating multiple models and various layer sizes, we found that concentrating FFNs in 70% of the consecutive layers around the middle of the Transformer LMs tends to yield superior performance for multiple downstream tasks compared with the baseline model using the standard FFN configuration. Interestingly, these results also suggest that FFNs in the first and last few layers may be redundant and that their functionality can be replaced by FFNs in the middle layers. These results suggest that an optimized model configuration exists other than simply placing FNNs evenly in each Transformer layer. We hope that the results of our experiments and our new findings will encourage further model analysis and the development of new Transformer LM configurations.

## Acknowledgments

This work was supported by the "R&D Hub Aimed at Ensuring Transparency and Reliability of Generative AI Models" project of the Ministry of Education, Culture, Sports, Science and Technology, and JST Moonshot R&D Grant Number JPMJMS2011-35 (fundamental research).

In this study, we mainly used ABCI 3.0 and the computer resource offered by Research Institute for Information Technology, Kyushu University under the category of General Projects. ABCI 3.0 is provided by AIST and AIST Solutions with support from "ABCI 3.0 Development Acceleration Use". Additionally, we partially used "mdx: a platform for building data-empowered society" for part of this research work.

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

# A  Detailed Model Configuration

|  | 285M, 12-layer | 570M, 24-layer | 570M, 40-layer | 1.2B, 40-layer |
|---|---|---|---|---|
| Layers | 12 | 24 | 40 | 40 |
| Hidden Dimension | 1280 | 1280 | 992 | 1440 |
| Intermediate Dimension | 4480 | 4480 | 3472 | 5040 |
| Attention Heads | 20 | 20 | 16 | 20 |
| Key/Value Heads | 20 | 20 | 16 | 20 |
| Activation Function | | SwiGLU | | |
| Vocabulary Size | | 50257 | | |

Table 3: **Hyperparameter Configurations for Model Architectures.**

# B  Intermediate Dimensions and Layer Placements of Experimental Models

The intermediate dimension $d'_f$ of FFN-expanded layers is recalculated based on the ratio of the FFN-expanded layers in each experimental model configuration and determined such that the total parameter count remains nearly identical to the baseline model. Table 4 shows the intermediate dimension $d'_f$ of FFN-expanded layers for each ratio of FFN-expanded layers. Note that when the ratio of FFN-expanded layers is 100%, the configuration is equivalent to the baseline model, so the dimensions correspond to those shown in Appendix A.

# C  Pretraining Settings

Table 5 presents the detailed pretraining configurations. To enable comprehensive exploration under computational resource constraints, this study conducted training with 20 times the number of tokens relative to model size, following the Chinchilla optimal (Hoffmann et al., 2022).

Regarding learning rate, for the 285M and 570M (24-layer) models, we adopted $3 \times 10^{-4}$, which is consistent with models reported in the literature including Pythia 410M (Biderman et al., 2023), OPT 350M (Zhang et al., 2022), Qwen 1.8B (Bai et al., 2023), OLMo 2 7B (OLMo et al., 2025), and Llama 3 8B (Grattafiori et al., 2024). For the 570M (40-layer) model, we initially experimented with $3 \times 10^{-4}$ but observed loss spikes and training instability in some experimental configurations. Since our study requires comprehensive comparisons across all configurations as shown in Figure 4, we adopted $1 \times 10^{-4}$ to ensure stable training across all settings for fair comparison. For the 1.2B (40-layer) model, we also set the learning rate to $1 \times 10^{-4}$ based on this observation.

Note that $1 \times 10^{-4}$ is not an extremely small value, as reference models such as Pythia 1.4B (Biderman et al., 2023) and OPT 1.3B (Zhang et al., 2022) use $2 \times 10^{-4}$, placing our choice within a reasonable range.

# D  Evaluation of Knowledge Capacity

Each instance in the Zero-Shot Relation Extraction (zsRE) dataset (Levy et al., 2017) consists of a knowledge-based question and its corresponding answer pair. In the task designed to measure knowledge capacity (zsRE task) (Mitchell et al., 2022; Cao et al., 2021), during evaluation, the model is provided either with only the question or with both the question and a portion of the answer and is tasked with generating the subsequent token.

Specifically, the process begins by prompting the model solely with the question to generate one token, which is then compared to the first token of the answer. Subsequently, the first token of the answer is appended to the original prompt to form a second prompt; the model

| Model | Ratio of FFN-expanded layers (%) | Intermediate dimension $d_f'$ |
|---|---|---|
| 285M, 12-layer | 10 | 53765 |
| | 30 | 17921 |
| | 50 | 8961 |
| | 70 | 6721 |
| | 90 | 5377 |
| 570M, 24-layer | 10 | 53765 |
| | 30 | 15361 |
| | 50 | 8961 |
| | 70 | 6721 |
| | 90 | 5121 |
| 570M, 40-layer | 10 | 34723 |
| | 30 | 11575 |
| | 50 | 6945 |
| | 70 | 4961 |
| | 90 | 3858 |
| 1.2B, 40-layer | 70 | 7201 |

Table 4: **Intermediate dimensions $d_f'$ of FFN-expanded Layers for Each Experimental Models.**

| | 285M, 12-layer | 570M, 24-layer | 570M, 40-layer | 1.2B, 40-layer |
|---|---|---|---|---|
| Global Batch Size | 288 | 560 | 560 | 1152 |
| Peak Learning Rate | $3 \times 10^{-4}$ | $3 \times 10^{-4}$ | $1 \times 10^{-4}$ | $1 \times 10^{-4}$ |
| Tokens | 5.8B | 11B | 11B | 23B |
| Laerning Rate Scheduler | | cosine | | |
| Sequence Length | | 1024 | | |
| Training Steps | | 20000 | | |
| Warmup | | 1000 | | |

Table 5: **Hyperparameter Settings for Pretraining.**

then generates one token, which is compared to the second token of the answer. This process is iterated until the entire answer has been generated.

The proportion of matching tokens computed relative to the complete answer is defined as the accuracy for that instance, and the average accuracy across all 19086 instances is then used as an indicator of the model's knowledge capacity.

# E  Baseline Model Performance and Literature Comparison

Since this study compares experimental models with baseline models using relative improvement (RI), we present the absolute performance of baseline models and comparison results with literature models to ensure the validity of RI-based discussions.

**Baseline Model Performance**    Table 6 shows the absolute performance of each baseline model on all downstream tasks used for evaluation as described in Section 4.3. As mentioned in Section 5, for the 285M and 570M model sizes, some of the baseline and experimental models did not achieve metric values above chance level (ARC-c: 0.25, Winogrande: 0.50) for ARC-c and Winogrande tasks, so we excluded them from our discussion for fair comparison.

**Comparison with Literature Models**    While direct performance comparison with literature models is challenging because many recent models employ extensive computational resources and often involve overtraining, we validated the appropriateness of our model performance using Pythia models (Biderman et al., 2023), which provide numerous intermediate checkpoints specifically for research purposes.

| Model | LAMBADA PPL | Wikitext PPL | ARC-c Acc | ARC-e Acc | HellaSwag Acc | LAMBADA Acc | PIQA Acc | WinoG Acc | zsRE Acc |
|---|---|---|---|---|---|---|---|---|---|
| baseline_285m_12l | 87.6 | **35.9** | **22.9** | **55.6** | **30.8** | 26.2 | **64.7** | 49.5 | **16.7** |
| pythia-410m-step3000 | **87.3** | 46.8 | 18.8 | 41.2 | 27.0 | **26.3** | 60.0 | **50.7** | 14.5 |
| baseline_570m_24l | 41.9 | **28.2** | **26.5** | **61.5** | **34.1** | 32.9 | **67.6** | 49.5 | **18.4** |
| baseline_570m_40l | 75.0 | 34.1 | 23.5 | 56.4 | 31.6 | 26.7 | 66.1 | **53.0** | 16.8 |
| pythia-1b-step5000 | **30.8** | 30.4 | 18.0 | 44.7 | 29.1 | **34.8** | 62.7 | 52.6 | 18.2 |
| baseline_1b_40l | 34.2 | 25.4 | **28.4** | **62.4** | **35.9** | 34.3 | **68.5** | **51.9** | 19.6 |
| pythia-1.4b-step11000 | **15.6** | **22.6** | 21.8 | 50.0 | 31.9 | **44.6** | 66.6 | 49.4 | **20.8** |

Table 6: **Absolute Performance of Baseline Models and Pythia Models.**

Specifically, we evaluated Pythia-410M (300M non-embedding parameters), Pythia-1B (806M non-embedding parameters), and Pythia-1.4B (1.2B non-embedding parameters) available on Hugging Face Hub, corresponding to our 285M, 570M, and 1.2B models, respectively, using checkpoints trained with equivalent token counts.

Table 6 presents the comparison results. Across all model sizes and evaluation tasks, our baseline models achieve performance equal to or superior to the literature models (Pythia). Even considering that the Pythia models used for comparison were intermediate checkpoints and may not have fully converged, these results clearly demonstrate that the performance of our baseline models under our training settings falls within a thoroughly reasonable range.

These results validate the reliability of our experimental findings and architectural comparisons, ensuring the validity of RI-based discussions.

## F  Additional Evaluation Results

Figure 6 presents the results across all downstream tasks not shown in Figure 4.

## G  Consistency of Results under Over-training Conditions

In this study, we conducted pre-training following the Chinchilla optimal (Hoffmann et al., 2022) with 20 times the number of tokens relative to model size to enable comprehensive exploration under computational resource constraints. However, many recent models employ over-training using large-scale computational resources. While over-training deviates from compute-optimal settings, it is known to potentially achieve higher performance improvements. Since our training configuration may be analyzing models at a stage where performance has not fully converged, it is necessary to verify the consistency of results under longer training periods.

Therefore, in this section, we validate that the main findings presented in Section 5 maintain consistency under over-training conditions. Due to computational resource constraints, we only focus on 285M (12-layer) and 1.2B (40-layer) models, conducting experiments under conditions with significantly extended training tokens for each of the baseline, first_70, middle_70, and final_70 configurations.

### G.1  Modifications to Experimental Settings

For this experiment, we modified the training configuration from Appendix C in the following two aspects:

**Total Training Tokens.**    For the 285M model, we set 8.8B tokens (approximately 20 times the model size of 413M including embedding and unembedding parameters) as 1× Chinchilla, and conducted training with 1×, 2× (17.6B tokens), 4× (35.2B tokens), and 8× (70.4B tokens). For the 1.2B model, we set 26B tokens (approximately 20 times the model size of 1.3B

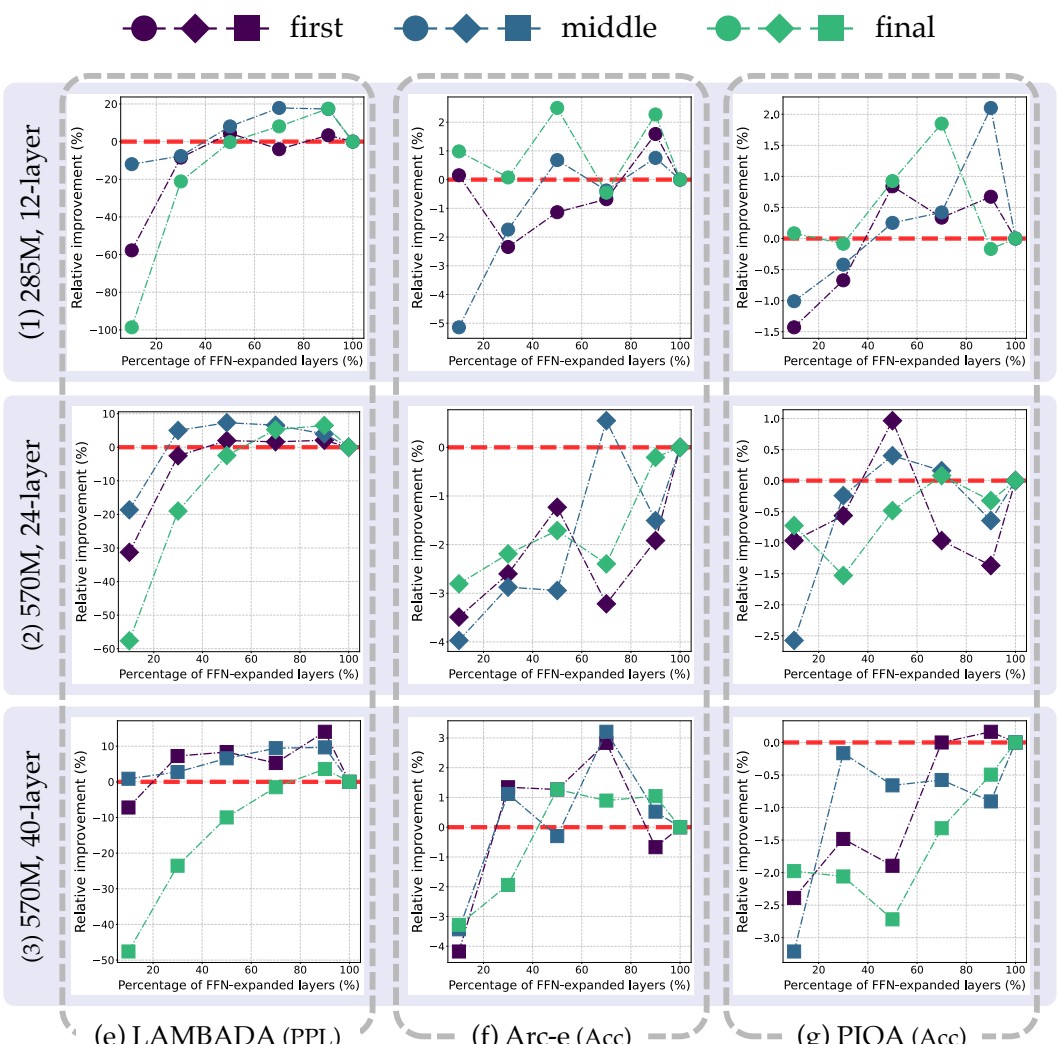

Figure 6: **Relative Improvement across Tasks by FFN-expanded Layer Ratio.** Relative improvement across tasks as a function of FFN-expanded layer ratio for different placement positions. Each row represents a different baseline configuration (model size and number of layers), while each column shows results for a different evaluation task. In each graph, the red dashed line highlights zero relative improvement, representing performance equivalent to the baseline model. Note that at 100% ratio, all configurations converge to the baseline performance regardless of placement position.

including embedding and unembedding parameters) as 1× Chinchilla, and conducted training with 1×, 2× (52B tokens), and 4× (104B tokens).

**Learning Rate Scheduler.** In this experiment, we employed the Warmup-Stable-Decay (WSD) scheduler (Hu et al., 2024) as the learning rate scheduler. The WSD scheduler maintains a constant learning rate for the majority of training and applies decay rapidly toward the end. A key advantage of this approach is the ability to resume training from checkpoints before the cooldown phase without changing the learning rate. Consequently, when extending training steps, there is no need to train from scratch, making this method highly efficient for over-training scenarios with excessive training steps.

Note that the models in the Section 5 used a cosine scheduler as described in Appendix C. To verify that the WSD scheduler functions appropriately, we compared validation loss

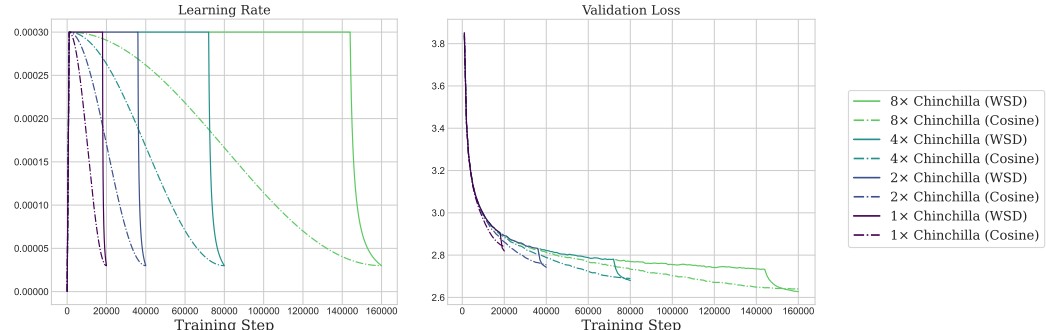

Figure 7: **Comparison of WSD and Cosine Schedulers.** Left panel shows learning rate curves, right panel shows validation loss curves.

| Model | 285M, 12-layer | | | | 1.2B, 40-layer | | |
| | 1× Chinchilla | 2× Chinchilla | 4× Chinchilla | 8× Chinchilla | 1× Chinchilla | 2× Chinchilla | 4× Chinchilla |
|---|---|---|---|---|---|---|---|
| first_70 | −1.08 | −0.99 | −0.51 | −2.59 | −1.39 | −1.41 | −2.13 |
| middle_70 | +3.48 | +3.12 | +4.28 | +3.18 | −0.70 | +0.09 | +0.04 |
| final_70 | +3.23 | +3.14 | +3.15 | +1.65 | −0.48 | −0.04 | +0.62 |

Table 7: **Average Relative Improvement (%) under Over-training Conditions.** Values represent performance averaged across the same downstream task sets used in Tables 1 for 285M (12-layer) and Table 2 for 1.2B (40-layer) models under different training token scales.

for the 285M baseline model with fixed warmup steps (1000) and peak learning rate (3e-4), using both cosine and WSD schedulers (Figure 7). The results confirmed that the WSD scheduler achieved lower loss than the cosine scheduler, validating that discussions can be conducted within an adequate performance range when adopting the WSD scheduler.

### G.2 Results

Table 7 presents the results of over-training experiments for each configuration. We confirmed that the main findings regarding the layerwise importance of FFNs remain consistent even under conditions with significantly increased training tokens. Specifically, examining Table 7, the middle_70 configuration demonstrates consistent advantages: for the 285M 12-layer model, it outperforms other configurations across all training token scales from 1× to 8× Chinchilla; for the 1.2B 40-layer model, while the performance gains are modest, the middle_70 configuration achieves positive average relative improvement from 2× to 4× Chinchilla training, indicating superior performance on downstream tasks compared to the baseline. These results demonstrate that our key insights remain robust even under conditions that exceed compute-optimal training settings.

## H Layerwise Importance Metric

Here, we provide the detailed formulation of the layerwise importance metric described in Section 5. For each layer $l$, we first calculate a raw importance score based on the performance impact when that layer's FFN is deactivated:

$$\text{Raw\_Importance}(l) = \frac{1}{C_l} \sum_{(p,r) \in \mathcal{S}} I_{(p,r)}(l) \tag{5}$$

$$I_{(p,r)}(l) = \begin{cases} -\frac{\text{RI}(p,r)}{|D_{(p,r)}|} & \text{if } l \in D_{(p,r)} \\ 0 & \text{otherwise} \end{cases} \tag{6}$$

where:

- $\text{RI}(p, r)$ is the average relative improvement across evaluation tasks for a configuration with position $p$ and ratio $r$
- $D_{(p,r)}$ is the set of FFN-deactivated layers in configuration $(p, r)$
- $|D_{(p,r)}|$ denotes the number of FFN-deactivated layers in configuration $(p, r)$
- $C_l$ is the number of configurations where layer $l$ was deactivated
- $\mathcal{S}$ represents all FFN placement configurations, defined by position $p \in \{\texttt{first}, \texttt{middle}, \texttt{final}\}$ and ratio $r \in \{10\%, 30\%, 50\%, 70\%, 90\%\}$

To facilitate comparison across different model sizes and configurations, we standardize these raw importance scores. Let $\mu$ and $\sigma$ be the mean and standard deviation of the raw importance scores across all layers. The final standardized importance score for each layer is given by:

$$\text{Importance}(l) = \frac{\text{Raw\_Importance}(l) - \mu}{\sigma} \tag{7}$$

This standardization ensures that the importance scores have zero mean and unit variance across all layers, making it easier to identify which layers contribute more or less than average to model performance. The standardized scores are used in the visualization presented in Figure 5, where positive values indicate layers with above-average importance and negative values indicate layers with below-average importance.

