# OpenReview forum: "Layerwise Importance Analysis of Feed-Forward Networks in Transformer-based Language Models"
_colmweb.org/COLM/2025/Conference — COLM 2025_

### Official Review · Reviewer_Nedb · 2025-05-05

**Rating:** 6
**Confidence:** 4
**Ethics Flag:** 1

**Summary:**

This paper makes an investigation into the importance of feed-forward layers (FFNs) in the multi-layer Transformer models, through allocating more parameters than others.  Through experiments on up to 1.2B parameter models up to 40 layers, the authors identified that FFNs in the middle to later layers are more beneficial to achieve higher downstream tasks, and thus more important than those in shallower layers.  While existing work is exclusively stacked the same shape of layers, this paper empirically shows the potential of searching for better shapes.

**Questions To Authors:**

- First of all, the paper does not explicitly state whether they are investigating encoders or decoders.  Readers can know they are decoder-only models only when they see "LLaMA" in line 65.

- Propose method modifies several layers.  Then, can the resulting models still be called "LLaMA architecture" (ll.70-71) ?

- How the authors come up with this "novel" approach (l.3)?  In other words, the rationale should be explained more precisely.

- The expanded width, i.e., $d_{f}'$, in the r% layers is determined on the basis of the number of (100-r)% of FFN-removal layers.  Please precisely describe that equation.  I'm wondering if the variants models can have exactly the same number of parameters.

- Accuracy and perplexity of the baseline models, i.e., the denominator for computing relative improvement (RI), are not shown, making it unclear if the RI is worth discussing and its scale.

- Is it possible to automatically identify the optimal (or at least reasonable) widths allocation?  This is especially important because the paper currently only calls more labor of future research.

- Some numbers in Section 5 are not corresponding to those in Figure 4; for example, +2.5% and +3.9% in l.209 look lower and higher in Figure 4 (1)(c) and Figure (1)(b), respectively.  +5.7% in l.224 looks lower than +5.0% line in Figure 4 (1)(a), and -1.8% in l.228 so does higher in Figure 4 (2)(d).  +8.5% in l.210 and %8.1% in l.232 do not match while they refer to the same point in Figure 4 (3)(b).

- Although it is commonly done, I'm not still sure if one can take "average" of different metric scores with different scales, as in Sections 5.3 and 5.4.  Could you give a justification?

- The superiority of "middle_70" for Wikitext, LAMBADA, HellaSwag, and zsRE is consistent with those in Table 1, and thus expected.  In contrast, how can the authors explain the largest performance drop in the new downstream tasks, i.e., ARC-c and Winogrande?

- The IDs in l.280 cover 9, which does not match 10 in l.283.  Similarly, mentions about layers in ll.298-302 often do not match with those in Figure 5.

**Reasons To Accept:**

- The paper brings to the community a new insight; stacking the same shape of layers is sub-optimal, and there can be a better way.

- Through a comprehensive comparison, although the search space is limited, the paper identified that FFNs in the middle layers are more important than those in the shallow and last layers, irrespective of the total numbers of layers.

- It is intriguing that FFNs for some layers can be completely removed.

**Reasons To Reject:**

- Experiments are not thorough.  For example, only two types of modifications, i.e., expansion and removal, have been proposed.  Why not narrow down but removal?  Why do not leave the original width at all? -> Would be taken down if the rationale of the choice of the two types is explained.

- The sizes of experimented models (with 285M to 1.2B parameters) are relatively small, compared to standard sizes of decoder-only models, and thus the paper does not guarantee that the findings can be applicable to much larger models, such as 80-layer and 126-layer models investigated under the name of Llama (70B and 405B models). -> Would be taken down considering the limitation of computes, but the generalizability of the findings to larger models should be discussed further.

- As listed below, the paper has several substantial issues in presentation.

---

> ### Author Response · Authors · 2025-06-03
>
> ### ## Answers to the Questions To Authors:
>
> > First of all, the paper does not explicitly state whether they are investigating encoders or decoders. Readers can know they are decoder-only models only when they see "LLaMA" in line 65.
>
> Thank you for pointing this out. As you stated, we mainly focus on decoder-only models. We thought that the main focus in the LM community recently has been that Transformer-based LMs are decoder-only models. We will refine our wording to clarify that our focus is on decoder-only models for this paper in the camera-ready version if our paper is accepted to the conference.
>
>
>
> > Propose method modifies several layers. Then, can the resulting models still be called "LLaMA architecture" (ll.70-71) ?
>
> Thank you for pointing this out as well.
> However, we would like to emphasize that we have already distinguished the LLaMA architecture from our modified models, which we refer to as experimental models.
> In other words, we did not intend to refer to our modified models as the “LLaMA architecture.”
>
> Please confirm that the reference in lines 70–71 only describes the baseline and the starting point of the experimental models.
> Subsequently, we consistently distinguish between baseline and experimental models using the “experimental model” or “[position][ratio][size]_[layers]” format throughout the paper.
>
> In any case, we will certainly refine our wording to avoid such confusion in the next version.
>
>
>
> > How the authors come up with this "novel" approach (l.3)? In other words, the rationale should be explained more precisely.
>
> Our approach is based on the observation from existing research that FFNs serve as a knowledge storage mechanism in Transformer LMs.
> This observation naturally raised questions for us: “Does the importance of FFNs depend on the layer position?” and “Can Transformer LMs properly train knowledge even by eliminating FFNs from certain layers?”
>
> From these questions, we aimed to empirically investigate the relationship between performance and the importance of FFN positions in Transformer LMs with unevenly located FFNs.
> Note that we attempted to directly verify FFN positional importance through architectural modification and training from scratch, unlike many previous studies that analyze pre-trained models without architecture modification.
>
>
>
> > The expanded width, i.e., df′, in the r% layers is determined on the basis of the number of (100-r)% of FFN-removal layers. Please precisely describe that equation. I'm wondering if the variants models can have exactly the same number of parameters.
>
> To maintain equal parameter counts, d'_f is calculated as: d'_f = d_f/r + (1-r)/(3r) ≈ d_f/r (when d_f is sufficiently large). We determine the intermediate dimension using the nearest integer value. This will be clarified in the camera-ready version if our paper is accepted to the conference.
>
>
>
> > Accuracy and perplexity of the baseline models, ... making it unclear if the RI is worth discussing and its scale.
>
> As noted in the footnote on page 5, tasks with performance below the chance level are excluded, ensuring the validity of the RI score. The baseline performance is available at: https://anonymous.4open.science/r/layer_wise_importance_of_ffn_for_rebuttal-3E06. We will include this in the camera-ready version.
>
>
> > Some numbers in Section 5 are not corresponding to those in Figure 4...
> > The IDs in l.280 cover 9, which does not match 10 in l.283. Similarly, mentions about layers in ll.298-302 often do not match with those in Figure 5.
>
> Thank you for identifying these inconsistencies. We will thoroughly verify all numerical correspondences between figures and text to ensure consistency in the camera-ready version.
>
>
>
> > Although it is commonly done, I'm not still sure if one can take "average" of different metric scores with different scales, as in Sections 5.3 and 5.4. Could you give a justification?
>
> Our justification includes: (1) Using relative improvement (RI) scores normalizes different scales, (2) Following established academic practices in language model research[1, 2, 3, 4], and (3) Confirming result robustness through consistent trends across individual tasks and averages.
>
> [1] Grattafiori et al. 2024, The Llama 3 Herd of Models
>
> [2] Raffel et al. 2019, Exploring the Limits of Transfer Learning with a Unified Text-to-Text Transformer
>
> [3] Chowdhery et al. 2022, PaLM: Scaling Language Modeling with Pathways
>
> [4] Le Scao et al. 2022, BLOOM: A 176B-Parameter Open-Access Multilingual Language Model
>
>
>
> > The superiority of "middle_70" for Wikitext, LAMBADA, HellaSwag, and zsRE ...the largest performance drop in the new downstream tasks, i.e., ARC-c and Winogrande?
>
> While training convergence is confirmed (see the provided link above), computational constraints limit the model size and training tokens. ARC-c and Winogrande require complex reasoning**, which** may be challenging under our limited training setup.

---

> ### Author Response · Authors · 2025-06-03
>
> We thank the reviewer for their valuable feedback and for taking the time to review our work. Please find our responses to each of your points below.
>
> ### ## 1. Experimental Scope and Design Choices
>
> > "Experiments are not thorough. For example, only two types of modifications, i.e., expansion and removal, have been proposed. Why not narrow down but removal? Why do not leave the original width at all?"
>
> We appreciate this insightful suggestion about exploring FFN width reduction rather than complete removal. This is indeed an interesting research direction that we did not explore due to the limited rebuttal period. If this paper is accepted, we will commit to investigating these additional configurations (including narrowed FFNs and retaining the original width) for the camera-ready version.)
> We acknowledge that the space of possible experimental configurations is vast, and we cannot exhaustively explore all variations within our resource constraints. However, within our defined scope, systematic exploration of FFN expansion and removal across multiple scales, positions, and ratios, our work provides meaningful contributions to understanding the layer-wise FFN importance. Our findings establish a foundation for future investigations, including the promising directions suggested by the reviewer.
>
> ### ## 2. The Size of Experimental Models
>
> > “The sizes of experimented models (with 285M to 1.2B parameters) are relatively small, … 80-layer and 126-layer models investigated under the name of Llama (70B and 405B models).”
>
> Our comprehensive study required substantial resources: 5,500+ GPU hours (H100) across 54 model configurations (285M-1.2B parameters). We followed a systematic approach: exhaustive small-scale exploration followed by targeted validation at larger scales.
> The consistent patterns across our 3× parameter range (285M → 1.2B) provide strong evidence for transferability.
>
> COLM's guidelines recognize that "most researchers do not have access to large-scale compute" and that limiting research to well-resourced labs "will stifle innovation." Our work exemplifies meaningful architectural insights within academic constraints. Training larger models during the rebuttal is practically impossible, requiring weeks of computation beyond available timelines.
>
> Our systematic exploration reveals novel layer-wise FFN importance patterns that provide specific hypotheses for future large-scale validation, rather than requiring complete re-exploration at larger scales.

---

> > ### Comment · Reviewer_Nedb · 2025-06-06
> >
> > Thank you very much for your response.
> >
> > I understand that the authors do not have enough computes to conduct thorough exploration of conceivable search spaces,
> > such as the types of of width modification and their exponential combinations, as well as the generalizability of the findings to larger models.
> > To be honest, I did not intend to request new results in the rebuttal, since it is for drawing proper review but not for updating the paper (after the submission deadline).
> >
> > I'd take down my first two "Reasons to Reject" but instead let me rephrase them into the following questions.
> >
> > 1. Among numerous ways of changing the width of feed-forward layers, what motivated the authors to begin with
> >    the two ways (expansion and whole-removal)?  Providing rational in the paper will be helpful to justify
> >    the authors' first choice in this new blanch of research.
> >
> > 2. As illustrated by Figure 5, "as models become deeper, FFNs may be more effective when placed relatively earlier in the network."
> >    This suggests that the same tendencies may be extrapolated for larger models, but the same allocation would not be optimal.
> >    Is it possible to automatically identify the optimal (or at least reasonable) widths allocation?
> >    This question is already posed but left unanswered.  Nevertheless, answer to this question is important as we can search only fewer configurations when investigating larger models.

---

> > > ### Author Response · Authors · 2025-06-07
> > >
> > > ### 1. Motivation for expansion and whole-removal
> > > > Among numerous ways of changing the width of feed-forward layers, what motivated the authors to begin with the two ways (expansion and whole-removal)? Providing rational in the paper will be helpful to justify the authors' first choice in this new branch of research.
> > >
> > > Our experimental design was motivated by the goal of structurally verifying, during pre-training, the existing hypothesis that FFNs function as knowledge storage devices. (Geva et al., 2021; Dai et al., 2022; Meng et al., 2022, as cited in Section 2 of our paper).
> > >
> > > **Fundamental motivation for removal operation**: Our primary operation is the "removal of FFNs" from certain layer(s). Our initial research question was whether FFNs in other layers could serve as alternative storage for the knowledge that would normally be stored by FFNs in specific layers under standard uniform FNN placement, in situations where FFNs in those layers are "absent." If such an alternation were impossible, then performance degradation should inevitably occur.
> > > Our results in Section 5 empirically showed that some configurations did not exhibit performance degradation, indicating that such alternation by FFNs in other layers is indeed possible.
> > >
> > > **Intent of combining with expansion operation**: Next, to further investigate FFNs as an alternative storage for the knowledge, we sought to examine whether concentrated knowledge accumulation is possible by redistributing the parameters lost through removal to FFNs in specific layers. As demonstrated in Table 1 (Section 5.3) and Table 2 (Section 5.4), we confirmed performance improvements for certain configurations, such as the middle 70, indicating that concentrating weights in FFNs of specific layers is effective for knowledge storage.
> > >
> > > **Rationale of experimental design**: Thus, we considered the combination of "removal" and "concentrated placement of removed weights" to be  appropriate operations for verifying in which layer positions FFNs can effectively store knowledge. As stated in Section 3.2, by maintaining all experimental models with exactly the same total number of parameters, we can measure pure placement effects.
> > >
> > >
> > >
> > > ### 2. Optimal Width Allocation for Larger Models
> > > > This suggests that the same tendencies may be extrapolated for larger models, but the same allocation would not be optimal. Is it possible to automatically identify the optimal (or at least reasonable) widths allocation?
> > >
> > > We appreciate the reviewer for highlighting that the optimal allocation of FFNs may vary depending on the number of layers. We understand the reviewer's concerns. We would like to address these concerns from the following two perspectives:
> > >
> > > First, although the number of parameters used in our experiments was relatively small, we validated models with depths of up to 40 layers. Considering that major current models such as LLaMA 8B have 32 layers [1], and Qwen3 14B has 40 layers [2], we believe we have covered a practical range in terms of layer count.
> > >
> > > Second, while full extrapolation to truly large (or deep) models remains challenging, we believe that our findings, specifically, concentrating FFNs in the middle to earlier layers, as indicated by the layer-wise importance in Fig. 5, represent an effective strategy that could be applied to larger (deeper) models. This is supported by the consistent trend observed across the 12-, 24-, and 40-layer models in our experiments.
> > >
> > > [1] Grattafiori et al. 2024, The Llama 3 Herd of Models
> > >
> > > [2] Yang et al. 2025, Qwen3 Technical Report

---

### Official Review · Reviewer_dJhd · 2025-05-11

**Rating:** 6
**Confidence:** 4
**Ethics Flag:** 1

**Summary:**

This paper investigates the importance of feed-forward networks (FFNs) in Transformers during pre-training. By studying different positioning and capacities given to FFNs (i.e. FFN-expanded and FFN-deactivated layers), the authors conclude that concentrating FFNs in 70% of the middle layers outperforms other configurations on multiple downstream tasks, including evenly placing FFNs in each Transformer layer.

**Questions To Authors:**

- What are the performances of the baseline architecture for the different benchmarks shown in Figure 4? The results are reported only in terms of relative improvement, but the performance at 0% improvement should also be reported for fair comparison with existing works.
- How do such performances compare with LLMs of similar size in the literature?
- What are the standard deviations across multiple runs for the reported results? Do the reported trends still hold in such a setting?
- Did you explore gradually increasing the capacity of the FFN layers until the middle layer, and then reducing it gradually?
- How can your findings be applied to MoE-based models?
- Did you experiment with adding depth instead of width for FFN expansion?
- Did you perform any hyperparameter tuning? In particular, how were the learning rates in Table 4 obtained?

**Reasons To Accept:**

- Studying the mechanisms behind Transformer components is an important research direction to promote safety and increase the overall understanding of Transformers models.
- The paper proposes a new configuration of FFNs in Transformers (as opposed to evenly placing them in every Transformer layer, as traditionally done) that works well across several downstream tasks.
- All models were pre-trained from scratch, ensuring a controlled environment.

**Reasons To Reject:**

- No reported standard deviation: The differences in performance between first, middle, and final configurations reported in Figure 4 are mostly overlapping, and it is hard to say that the trends are statistically significant without performing multiple runs and reporting standard deviations. This also applies to the results reported in Tables 1 and 2.
- Only one architecture was used throughout the study, for which the authors state that "the LLaMA architecture has become the de facto standard model". In my opinion, this claim should be edited to avoid confusion since there is currently a clear trend toward mixture-of-experts (MoE) architectures (even Llama-4 now adopts MoE). I understand that there are limited resources, but the authors could have investigated additional base architectures such as the one used in GPT-2 (i.e. LayerNorm and GELU).
- Small-scale experiments: I understand that access to computing resources may be limited, but 1.2B parameters is still considered a very small model in today's terms.
- No hyperparameter tuning: As I understand it, there is no mention of any hyperparameter tuning for pre-training. Even though this can be related to limited resources, there are efficient alternatives to optimize hyperparameters on small proxy models (e.g. maximal update parameterization) or by simply training for a reduced number of iterations and resuming the best hyperparameter configurations.
- The FFN-expansion method could have been further investigated. For example, regarding whether to add depth instead of width to the network. The authors claim that the "extreme concentration of FFN parameters in very few layers compromises model capability, likely because reducing the layers applying non-linear transformations limits representational capacity". Adding FFN layers interleaved with non-linearities could have been done to further study the impact of having fewer or more non-linearities while maintaining the same number of parameters as the width-based expansion proposed in the paper.

---

> ### Author Response · Authors · 2025-06-03
>
> We thank the reviewer for their valuable feedback and for taking the time to review our work. Please find our responses to each of your points below.
>
> ## ## 1. Standard Deviation and Statistical Significance
> We understand the importance of conducting multiple training runs with different seeds when evaluating different architectures, but this is not practically feasible due to computational resource constraints.
> As a general approach, numerous strong papers have adopted the method of conducting single training runs with fixed seeds and demonstrating model superiority by evaluating on more tasks [1, 2, 3, 4]. The downstream task evaluations in these papers use greedy sampling strategies and do not output standard deviations from multiple trials. Therefore, we are evaluating our method in an academically justified setting.
>
>
> [1] Grattafiori et al. 2024, The Llama 3 Herd of Models
>
> [2] Raffel et al. 2019, Exploring the Limits of Transfer Learning with a Unified Text-to-Text Transformer
>
> [3] Chowdhery et al. 2022, PaLM: Scaling Language Modeling with Pathways
>
> [4] Le Scao et al. 2022, BLOOM: A 176B-Parameter Open-Access Multilingual Language Model
>
> ## ## 2. Insufficient Diversity in Base Architecture
> > "Only one architecture was used throughout the study... there is currently a clear trend toward mixture-of-experts (MoE) architectures"
>
> We understand the reviewer's observation regarding the development of MoE architectures. However, we respectfully maintain that LLaMA represents the most appropriate choice for our research objectives, though we acknowledge the need to clarify our "de facto standard" characterization.
> Rationale for LLaMA selection: Our study aims to isolate and understand the fundamental mechanisms of FFN layer importance. This requires a controlled experimental environment where we can attribute performance changes specifically to FFN positional modifications rather than confounding architectural factors. LLaMA provides this foundation through:
> Integration of modern improvements (Pre-LN, SwiGLU, RoPE) ensuring relevance to contemporary models
> Extensive academic validation and reproducible implementations
> Well-understood component behaviors that allow clean isolation of FFN effects
> Comparison with alternative architectures: While we appreciate the suggestion to explore GPT-2 (LayerNorm + GELU), its lack of modern architectural improvements would compromise the relevance of our findings to current model development. Similarly, MoE architectures, despite their growing importance, introduce additional complexity through expert selection mechanisms and sparse activation patterns that would confound our ability to measure pure FFN positional effects.
>
>
>
> ## ## 3. Insufficient Exploration of FFN Expansion Methods
> > "The FFN-expansion method could have been further investigated. For example, regarding whether to add depth instead of width to the network."
>
> Our research scope is specifically analytical, aiming to verify FFN importance within existing architectures. As stated in lines 37-38, our research question (RQ) focuses on understanding "the roles and functions of FFNs within existing Transformer-LMs, particularly how their importance varies by layer position."
> While adding depth is an important perspective for architectural design, it addresses a fundamentally different problem from our RQ for the following reasons:
> Baseline comparison validity: Adding depth fundamentally alters the architecture itself, changing our analytical target of "FFN importance within existing Transformer-LMs."
>  Causal confounding: Depth addition conflates "positional importance changes" with "structural performance changes," preventing pure isolation of the layer position effects we aim to understand.
> In summary, depth expansion falls outside our research scope as it pertains to architectural design rather than our analytical investigation of FFN mechanisms within established Transformer structures.

---

> > ### Comment · Reviewer_dJhd · 2025-06-07
> >
> > I acknowledge the comprehensive response and would like to thank the authors for their time. While some of my questions and concerns were addressed, other questions remain unanswered:
> >
> > > What are the performances of the baseline architecture for the different benchmarks shown in Figure 4? The results are reported only in terms of relative improvement, but the performance at 0% improvement should also be reported for fair comparison with existing works.
> >
> > > How do such performances compare with LLMs of similar size in the literature?
> >
> > > Did you perform any hyperparameter tuning? In particular, how were the learning rates in Table 4 obtained?
> >
> > I would appreciate having responses to these before making my final recommendation. Thank you in advance.

---

> > > ### Author Response · Authors · 2025-06-09
> > >
> > > ### The performance of the baseline architecture
> > > > What are the performances of the baseline architecture for the different benchmarks shown in Figure 4? The results are reported only in terms of relative improvement, but the performance at 0% improvement should also be reported for fair comparison with existing works.
> > >
> > > As noted in the footnote on page 5, tasks with performance below chance level on the baseline models were excluded to ensure the validity of the RI score. Please do not misunderstand: we did not exclude tasks based on RI scores showing 0% improvement or degradation in the proposed models. We excluded tasks solely based on baseline performance. Therefore, we believe our evaluation was conducted under fair conditions.
> > >
> > > The baseline performance is available at: https://anonymous.4open.science/r/layer_wise_importance_of_ffn_for_rebuttal-3E06. We will include this in the camera-ready version.
> > >
> > > ### Comparison with Literature Models
> > > > How do such performances compare with LLMs of similar size in the literature?
> > >
> > > We appreciate the reviewer's question regarding how our model performances compare with similar-sized LLMs in the literature. We address this concern through two key aspects:
> > >
> > >
> > > **Rationale for Chinchilla Rule Adoption** : In this study, we applied the Chinchilla rule [1] for determining training token counts, training our models with 20× the number of tokens relative to model size. This approach enables comprehensive exploration within our limited computational resources. As evidenced by the training and validation loss curves provided in the supplementary materials at: https://anonymous.4open.science/r/layer_wise_importance_of_ffn_for_rebuttal-3E06, our models demonstrate reasonable convergence, indicating that the token count is sufficient for fair evaluation rather than being inadequately trained.
> > >
> > > **Validation of Model Performance Under Our Training Settings** : While many recent models employ massive computational resources and often involve extensive overtraining, making direct performance comparisons with literature models challenging, we validate the appropriateness of our model performance using Pythia models [2], which provide numerous intermediate checkpoints specifically for research purposes.
> > > Specifically, we evaluated Pythia models available on Hugging Face Hub - namely Pythia-410M (300M non-embedding parameters), Pythia-1B (806M non-embedding parameters), and Pythia-1.4B (1.2B non-embedding parameters) - using our evaluation pipeline. These correspond to our 285M, 570M, and 1B models, respectively. We selected checkpoints that had been trained on token counts equivalent to our corresponding models.
> > > The comparison results are presented in the table below. Across all model sizes and evaluation tasks, our baseline models achieve performance equal to or superior to the literature models (Pythia). We acknowledge that the Pythia models used for comparison were intermediate checkpoints and thus may not have fully converged during their original training. This clearly demonstrates that the performance of our baseline models under our training settings falls within a thoroughly reasonable range, validating the reliability of our experimental findings and architectural comparisons.
> > >
> > > | model_name | ARC-c,acc | ARC-e,acc | HellaSwag,acc | LAMBADA,acc | OBQA,acc | PIQA,acc | Wikitext,ppl | Winogrande,acc | zsRE,acc |
> > > | --- | --- | --- | --- | --- | --- | --- | --- | --- | --- |
> > > | baseline_285m_12l | 22.9 | 55.6 | 30.8 | 26.2 | 20.6 | 64.7 | 35.9 | 49.5 | 16.7 |
> > > | pythia-410m-step3000 | 18.8 | 41.2 | 27 | 26.3 | 15.4 | 60 | 46.8 | 50.7 | 14.5 |
> > > | baseline_570m_24l | 26.5 | 61.5 | 34.1 | 32.9 | 22.4 | 67.6 | 28.2 | 49.5 | 18.4 |
> > > | baseline_570m_40l | 23.5 | 56.4 | 31.6 | 26.7 | 20 | 66.1 | 34.1 | 53 | 16.8 |
> > > | pythia-1b-step5000 | 18.0 | 44.7 | 29.1 | 34.8 | 17.4 | 62.7 | 30.4 | 52.6 | 18.2 |
> > > | baseline_1b_40l | 28.4 | 62.4 | 35.9 | 34.3 | 23.6 | 68.5 | 25.4 | 51.9 | 19.6 |
> > > | pythia-1.4b-step11000 | 21.8 | 50.0 | 31.9 | 44.6 | 29.6 | 66.6 | 22.6 | 49.4 | 20.8 |

---

> > > > ### Author Response · Authors · 2025-06-09
> > > >
> > > > ### Learning Rate Selection
> > > > > Did you perform any hyperparameter tuning? In particular, how were the learning rates in Table 4 obtained?
> > > >
> > > >
> > > > We acknowledge that due to limited computational resources, we did not perform comprehensive hyperparameter tuning across all experimental configurations. However, the learning rates for each setting were determined based on the following rationale:
> > > >
> > > > For the 285M and 570M (24-layer) models, we set the learning rate to 3e-4 based on models in the literature (Pythia 410M [2], OPT 350M [3], Qwen 1.8B[4], OLMo 2 7B[5], Llama 3 8B[6]) that adopt the same rate.
> > > >
> > > > For the 570M (40-layer) model, we initially experimented with 3e-4 but observed loss spikes and training instability in some experimental configurations. Since our study requires comprehensive comparisons across all configurations as shown in Figure 4, we adopted 1e-4 to ensure stable training across all settings for fair comparison. For the 1.2B (40-layer) model, we set the learning rate to 1e-4 based on this observation.
> > > >
> > > > Note that 1e-4 is not an extremely small value, as reference models such as Pythia 1.4B [2] and OPT 1.3B [3] use 2e-4, placing our choice within a reasonable range. For models with many layers, gradient propagation characteristics change, requiring more conservative learning rate settings for stable training.
> > > >
> > > > [1] Hoffmann et al. 2022, Training Compute-Optimal Large Language Models
> > > >
> > > > [2] Biderman et al. 2023, Pythia: A Suite for Analyzing Large Language Models Across Training and Scaling
> > > >
> > > > [3] Zhang et al. 2022, OPT: Open Pre-trained Transformer Language Models
> > > >
> > > > [4] Bai et al. 2023, Qwen Technical Report
> > > >
> > > > [5] Walsh et al. 2024, 2 OLMo 2 Furious
> > > >
> > > > [6] Grattafiori et al. 2024, The Llama 3 Herd of Models

---

> > > > > ### Comment · Reviewer_dJhd · 2025-06-11
> > > > >
> > > > > Thank you for providing clarifications to my unanswered questions, which have now been addressed. I have updated my final score from 5 to 6.

---

> ### Author Response · Authors · 2025-06-03
>
> ## ## 4. The Size of Experimental Models
> > "Small-scale experiments: I understand that access to computing resources may be limited, but 1.2B parameters is still considered a very small model in today's terms."
>
> Our comprehensive study required substantial resources: 5,500+ GPU hours (H100) across 54 model configurations (285M-1.2B parameters). We followed a systematic approach: exhaustive small-scale exploration followed by targeted validation at larger scales.
> The consistent patterns across our 3× parameter range (285M → 1.2B) provide strong evidence for transferability.
>
> COLM's guidelines recognize that "most researchers do not have access to large-scale compute" and that limiting research to well-resourced labs "will stifle innovation." Our work exemplifies meaningful architectural insights within academic constraints. Training larger models during the rebuttal is practically impossible, requiring weeks of computation beyond available timelines.
>
> Our systematic exploration reveals novel layer-wise FFN importance patterns that provide specific hypotheses for future large-scale validation, rather than requiring complete re-exploration at larger scales.

---

### Official Review · Reviewer_XrFb · 2025-05-13

**Rating:** 7
**Confidence:** 4
**Ethics Flag:** 1

**Summary:**

This paper concentrate on the niche topic of FFN layer importance, given a compute budget. The author devices an effective searching strategy to decide FFN. The paper is mostly clearly written after one understand the notation of different models configuration. The experiment and evaluation setup are relatively solid and comprehensive for them to make claims.

The down side of the work is that it focus much on a niche topic; and it might be hard to connect it to a boarder use case like "how would the message in the paper help people do neural architecture search?" or "what's the next neural architecture?".

**Questions To Authors:**

1. Hypothetically, do you think using a different search heuristics will arrive at the same conclusion?
2. Figure 3, how could the percentage be 90%? if 90% FFN is expanded, where does the author reduce the parameter to make sure the compute budget is the same?
> when the ratio of FFN-expanded layers is 100%, the model architecture is identical to the baseline

I am confused that an expansion ratio of 100% is equal to baseline (no expansion). I felt the largest ratio one could do is 50%. Also, what's the strategy to remove layers? One possible explanation is: the intermediate size d'_f of FF-expanded is recalculated per ratio of  FFN-expanded layers. If this is the case that makes sense.
3. Figure 5 axis label are not visible.

**Reasons To Accept:**

1. Design effective searching strategies to decide on which FFN to increase size or decrease size.
2. Found interesting observation that concentrating FFN on the 70% of middle layers will lead to best performance; and observe that this configuration holds across scale

**Reasons To Reject:**

1. The authors assumes FFN-expanded are equally important. However, this likely might not be the case as reflected in Figure 5. For example, could the author redistribute the size of FFN-expended on the "blue" layers according to Figure 5, and obtain even better performance?
2. Have the author used techniques from [1], would the layer importance conclusion differ?

[1] Studying Large Language Model Generalization with Influence Functions

---

> ### Author Response · Authors · 2025-06-03
>
> We thank the reviewer for their valuable feedback and for taking the time to review our work. Please find our responses to each of your points below.
>
>
> ## ##1. Contribution to Practicality and Architecture Search
>
> > The down side of the work is that it focus much on a niche topic; and it might be hard to connect it to a boarder use case like "how would the message in the paper help people do neural architecture search?" or "what's the next neural architecture?".
>
> This research addresses an important and practical topic for the following reasons:
> FFNs are a core component because they occupy approximately 2/3 of the parameters in transformers, and their optimization directly impacts overall model performance.
> From this perspective, we believe that the research on FFN placement strategies is not niche but rather one of the most critical research topics for improving transformers.
>
> Practical value of this research: fundamental questioning of existing paradigms – we empirically demonstrated that the currently mainstream uniform placement across all layers may not be optimal.
>
> New direction for architecture search: By showing that a middle 70% placement consistently achieves superior performance across multiple model sizes, we have opened up a new dimension of Neural Architecture Search involving non-uniform FFN parameter strategies.
>
>
>
> ## ##2. Assumption of Equal Importance for FFN-Expanded Layers
>
> > The authors assumes FFN-expanded are equally important. However, this likely might not be the case as reflected in Figure 5.
>
> The "equal importance" assumption you pointed out is not a claim that FFN-expanded layers are actually equally important, but rather a methodological starting point we adopted because directly quantifying the individual contribution of each layer in specific configurations is technically extremely challenging.
> Interestingly, despite this simplified approach, we observed a clear trend of consistently higher importance for middle layers across multiple model configurations, suggesting that our analytical method can significantly capture actual layer-wise importance differences.
> Your suggestion of non-uniform allocation based on importance to the "blue" layers in Figure 5 is a natural and important next step for this research. Allocating more parameters to layers with higher importance scores could be expected to yield further performance improvements, making this a valuable research direction to pursue.
>
>
>
> ## ##3. Relationship with Research Using Influence Functions
>
> > Have the author used techniques from [1], would the layer importance conclusion differ? [1] Studying Large Language Model Generalization with Influence Functions
>
> The paper [1] you mentioned employs a post-hoc analytical approach that examines the layer-wise influence of training data on already pre-trained models with standard Transformer architectures.
>
> In contrast, our research takes a fundamentally different approach by modifying the model architecture itself from the beginning of training, such as removing FFNs from certain layers during the pre-training stage. Since [1] analyzes standard architectures after training completion while our work involves architectural modifications during training, the methodologies are incompatible and orthogonal.
>
>
>
> ## ##4. The Method for Determining Intermediate Dimensions of FFNs in Experimental Models
> > Figure 3, how could the percentage be 90%?
>
> As you mentioned, d'_f is indeed recalculated based on the proportion of FFN-expanded layers. Specifically, when 90% of layers are FFN-expanded and 10% are FFN-deactivated (completely removed), we recalculate the intermediate dimension d'_f of FFN-expanded layers according to the ratio r to maintain the same total parameter count. This mechanism ensures that at 100% ratio, all layers have standard FFNs, making the model identical to the baseline.
>
>
>
> ## ##5. The Axis Labels in Figure 5
>
> > Figure 5 axis label are not visible.
>
> Thank you for your feedback. In Figure 5, the horizontal axis represents layer indices, and the vertical axis represents the layer-wise importance scores (standardized) calculated according to Appendix D. We will improve the visibility issues in the camera-ready version.

---

> > ### Comment · Reviewer_XrFb · 2025-06-07
> >
> > Thanks for the author in addressing my concern.

---

### Official Review · Reviewer_8WSS · 2025-05-13

**Rating:** 5
**Confidence:** 3
**Ethics Flag:** 1

**Summary:**

This paper is addressing a very fundamental and interesting problem of identifying how does the placement and the properties of FFN influences the model's overall performance. This is interesting as most of the computation and params are allocated to FFNs and not to self attention blocks. The authors have evaluated transformer-LMs which they they trained from scratch with various FFN configurations.

**Questions To Authors:**

1) When FFNs are expanded how attention behaves?

2) Why in case of smaller models 285M (Table 1) first layer expansion outperforms the middle layer expansion?

**Reasons To Accept:**

1) This paper studies an important and under studied problem.

2) The experiments are reasonably large scale and lots of empirical results are shown.

3) The results look strong and although the scaling law is not what is explicitly mentioned but it seems the best expanded configurations should work in large scale as well.

**Reasons To Reject:**

1) The takeways are not clearly mentioned, and the writing is a bit weak.

2) Since the authors are training the model from scratch can they show the impact of the best configuration (i.e. middle layer expansion) on pre-train val loss/ train loss. How the convergence and generalization looks like in these scenarios.

3) The results in Fig 4 are not conclusive as the gains are highly dependent on the dataset.

4) Not much intuition is discussed through out the paper.

---

> ### Author Response · Authors · 2025-06-03
>
> We thank the reviewer for their valuable feedback and for taking the time to review our work. Please find our responses to each of your points below.
>
> ## ## 1.
>
> > "The takeways are not clearly mentioned, and the writing is a bit weak."
>
> We respectfully disagree with the observation that the takeaways (our main contributions) are unclear. Our primary finding, namely, concentrating FFNs in 70% of consecutive intermediate layers while maintaining the same parameter count consistently outperforms standard configurations across multiple model sizes and tasks, is clearly articulated throughout the paper in the following locations: Abstract (lines 10-14), Section 5.2 (lines 233-235), Section 5.3 (lines 246-248), Section 5.4 (lines 262-267), and Conclusion (lines 310-315).
>
>
>
> ## ## 2.
>
> > "Since the authors are training the model from scratch can they show the impact of the best configuration (i.e. middle layer expansion) on pre-train val loss/ train loss. How the convergence and generalization looks like in these scenarios."
>
> The following link shows the training and validation learning curves:  https://anonymous.4open.science/r/layer_wise_importance_of_ffn_for_rebuttal-3E06.
> We will provide all the logs for training and evaluation from our experiments once this paper is accepted to the conference.
>
>
>
> ## ## 3.
>
> > "The results in Fig 4 are not conclusive as the gains are highly dependent on the dataset."
>
> We acknowledge that the gains depend on the task, but this variation is unavoidable for the following reasons:
> - **Dependence on Task Difficulty** : The extent of improvement naturally hinges on the complexity of each task—easier tasks tend to show smaller performance boosts. Since it's fundamentally impossible to objectively normalize task difficulty across different benchmarks, it's unrealistic to expect consistent gains for all tasks.
> - **Deliberate Evaluation Design** : We've tackled this issue through meticulous experimental planning:
>   - Section 4.3 details evaluations over a broad range of task types, including language modeling, commonsense reasoning, and knowledge evaluation.
>   - As mentioned on page 5, we excluded tasks where the baseline model performed worse than chance.
>   - We also verified consistent trends across different model sizes (285M, 570M, 1.2B) and layer counts (12, 24, 40).
> - **Reliable Patterns** : Even though gains differ by task, we observed stable patterns across tasks and model configurations in Sections 5.1 and 5.2:
>   - Low FFN expansion ratios (10–30%) consistently underperform.
>   - High ratios (70–90%) often surpass the baseline.
>   - Middle placement of FFN layers tends to outperform initial and final placements.
>  These consistently observed trends across varied conditions strongly support our conclusions. Therefore, while Fig. 4 reflects task-related variability, it doesn’t weaken the credibility of our findings.
>
>
>
> ## ## 4.
> > "Not much intuition is discussed through out the paper."
>
> An intuitive explanation of why the “Middle-70” configuration is effective:
>
> - **Theoretical background** : Previous studies have demonstrated that the most significant information processing for downstream tasks primarily occurs in the mid-to-final FFN layers of a model [1,2].
>
> - **Effectiveness of Middle-70** : By concentrating FFN layers in the parts of the model that matter most for downstream tasks, the parameter budget is used more efficiently.
>
> - **Novelty** : Unlike prior work that analyzed publicly released pre-trained models (static models), we trained models from scratch with different FFN settings in specific layers, yet still achieved performance equal to or better than the baseline. This reveals an intriguing aspect of the Transformer’s redundancy and adaptability.
>
> - **Real-world implications** : This approach offers a promising new direction for designing efficient models that deliver higher performance within the same parameter constraints.
>
> - [1] Meng et al. 2022, Locating and Editing Factual Associations in GPT
> - [2] Geva et al. 2021, Transformer Feed-Forward Layers Are Key-Value Memories
>
>
>
> ## ## 5.
>
> > "Why in case of smaller models 285M (Table 1) first layer expansion outperforms the middle layer expansion?"
>
> I would like to clarify a misunderstanding. It is not that the first layer expansion outperforms other configurations; rather, “middle 70” ranked third, surpassing the baseline by 2.14%, which supports our claim. Furthermore, in both the 570M (24-layer and 40-layer) models, “middle 70” achieved first place. Its superiority becomes even more apparent as the number of layers increases. This is likely because, as the layer count rises, the overlap between layers designated for FFN expansion in the first, middle, and final settings diminishes, causing the performance differences to appear more sharply.
>
> We appreciate your thoughtful comments and believe that our responses clarify the contributions of our work while also outlining promising directions for future research.

---

> > ### Comment · Reviewer_8WSS · 2025-06-08
> > **Rebuttal Acknowledgement**
> >
> > Dear Authors,
> >
> > Thanks for responding to questions and concerns. I acknowledge that I have read the author response and also the new results shared via anonymous link. The difference between the training trajectories of the baseline and middle-70 is very small. I am keeping the score.

---

> > ### Author Response · Authors · 2025-06-09
> >
> > > The difference between the training trajectories of the baseline and middle-70 is very small. I am keeping the score.
> >
> > We appreciate the reviewer’s acknowledgement. However, we must respectfully disagree with keeping the score primarily based on the training trajectories, as mentioned in the reviewer’s previous comment.
> >
> > Recent research has clearly suggested that the training trajectory does not always correlate with downstream task performance [1, 2, 3, 4]. We should use the training trajectory only to evaluate the appropriateness and robustness of training within a single model, and not to compare different models. Furthermore, as demonstrated by major LLM developments [5, 6, 7, 8], the overall performance of LLMs has been evaluated across a diverse range of downstream tasks, and we have followed this approach.
> >
> >
> > In line with this modern evaluation paradigm, our paper explicitly shows that our best configuration ("middle 70") consistently outperforms the baseline across multiple tasks, model sizes, and layer counts. This strong and consistent improvement across diverse downstream tasks, such as LAMBADA, Wikitext, HellaSwag, and zsRE, represents a highly significant finding. These comprehensive evaluations robustly demonstrate the effectiveness of our architectural modifications, providing much stronger evidence than training trajectory alone.
> >
> > [1] Liu et al. 2022, Same Pre-training Loss, Better Downstream: Implicit Bias Matters for Language Models
> >
> > [2] Kaddour et al. 2023, Challenges and Applications of Large Language Models
> >
> > [3] Levy et al. 2024, Same Task, More Tokens: the Impact of Input Length on the Reasoning Performance of Large Language Models
> >
> > [4] Tay et al. 2022, Scaling Laws vs Model Architectures: How does Inductive Bias Influence Scaling?
> >
> > [5] Grattafiori et al. 2024, The Llama 3 Herd of Models
> >
> > [6] Anil et al. 2023, Gemini: A Family of Highly Capable Multimodal Models
> >
> > [7] Jiang et al. 2023, Mistral 7B
> >
> > [8] Yang et al. 2025, Qwen3 Technical Report

---

### Decision · Program_Chairs · 2025-07-08

**Decision:**

Accept

**Comment:**

The paper investigates the role and importance of feed-forward network (FFN) layers within Transformer language models, focusing on how different placements and sizes of FFNs affect overall model performance. The authors conduct experiments by training Transformer-based language models from scratch with varying FFN configurations, including expanding or removing FFNs in specific layers. Their empirical results suggest that concentrating FFN capacity in approximately 70% of the middle layers yields better performance across multiple downstream tasks compared to evenly distributing FFNs across all layers. The study explores both FFN expansion and removal to find more optimal architectural configurations rather than uniformly sized layers. The goal is to provide insights into more effective parameter allocation within Transformer models to improve efficiency and accuracy.

Pros:
- The paper addresses an underexplored aspect of Transformer architecture related to FFN placement and sizing.
- It provides extensive empirical evaluation with experiments on models trained from scratch.
- The findings show that concentrating FFNs in the middle layers improves performance across multiple tasks.
- The study challenges the standard practice of evenly sized FFNs, offering new insights into efficient parameter allocation.
- The experiments include both FFN expansion and removal, broadening the scope of architectural exploration.

Cons:
- Experiments are limited to relatively small model sizes and a single base architecture.
- There is no reported standard deviation or multiple runs to assess statistical significance.
- The impact of FFN expansion on attention behavior is not analyzed.
- Hyperparameter tuning and exploration of alternative FFN modifications, such as adding depth, are missing.
- Some results are dataset-dependent and not fully conclusive.
- Reviewers complain that the presentation contains inconsistencies and unclear explanations in parts.

The meta-reviewer notes that the contents of the reviews are somewhat misaligned (i.e., more negative) with the final scores in some cases.